# Effects of an App-Based Intervention Program to Reduce Substance Use, Gambling, and Digital Media Use in Adolescents and Young Adults: A Multicenter, Cluster-Randomized Controlled Trial in Vocational Schools in Germany

**DOI:** 10.3390/ijerph20031970

**Published:** 2023-01-20

**Authors:** Benjamin Pietsch, Nicolas Arnaud, Kirsten Lochbühler, Monika Rossa, Ludwig Kraus, Elena Gomes de Matos, Kristin Grahlher, Rainer Thomasius, Reiner Hanewinkel, Matthis Morgenstern

**Affiliations:** 1IFT-Nord Institute for Therapy and Health Research, 24114 Kiel, Germany; 2German Centre for Addiction Research in Childhood and Adolescence, University Medical Centre, Hamburg-Eppendorf, 20251 Hamburg, Germany; 3IFT Institut für Therapieforschung, 80804 Munich, Germany; 4Department of Public Health Sciences, Centre for Social Research on Alcohol and Drugs, Stockholm University, 11419 Stockholm, Sweden; 5Institute of Psychology, ELTE Eötvös Loránd University, 1053 Budapest, Hungary; 6Institute for Medical Psychology and Medical Sociology, University Medical Center Schleswig-Holstein (UKSH), 24105 Kiel, Germany

**Keywords:** prevention, vocational students, voluntary commitment, abstinence, substance use, internet-related problems, cluster-randomized controlled trial

## Abstract

Vocational students are a risk group for problematic substance use and addictive behaviors. The study aim was to evaluate the effects of an app-based intervention on tobacco, e-cigarettes, alcohol, and cannabis use as well as gambling and digital media-related behaviors in the vocational school setting. A total of 277 classes with 4591 students (mean age 19.2 years) were consecutively recruited and randomized into an intervention (IG) or waitlist control group (CG). Students from IG classes received access to an app, which encouraged a voluntary commitment to reduce or completely abstain from the use of a specific substance, gambling, or media-related habit for 2 weeks. Substance use, gambling, and digital media use were assessed before and after the intervention in both groups with a mean of 7.7 weeks between assessments. Multi-level logistic regression models were used to test group differences. Intention-to-treat-results indicated that students from IG classes had a significantly larger improvement on a general adverse health behavior measure compared to CG (OR = 1.24, *p* = 0.010). This difference was mainly due to a significantly higher reduction of students’ social media use in the IG (OR = 1.31, *p* < 0.001). Results indicate that the app “Meine Zeit ohne” is feasible for the target group and seems to have a small but measurable impact on students’ health behavior.

## 1. Introduction

There is a substantial increase in substance use (tobacco, e-cigarettes, cannabis, alcohol) in the trajectory from adolescence to early adulthood [1,2,3,4]. This period of life is accompanied by a number of critical life decisions regarding one‘s personal, educational, and vocational future; the need to take more responsibility for one’s own actions; and often by a change of the home and educational setting [2]. Students in German vocational schools receive practice-oriented education while being employed and working as apprentices in a company. Most of them are aged 16 to 20 years and visit vocation schools for 2 or 3 years. There is evidence for higher substance use rates among students at vocational schools compared to their peers in the general population [1,5,6]. For example, in one study, 62% of German vocational students reported to smoke frequently, 45% reported a risky pattern of alcohol use, and 4% were problem cannabis users according to the Severity of Dependence Scale [6]. Research suggests that alcohol, tobacco, cannabis, and even other drugs like cocaine or benzodiazepines are often used to relieve work-related stress [7,8]. It seems plausible that this is also true for vocational students who might use substances to relieve training-related stress, over and above other factors that contribute to increased substance use in this age group (e.g., self-exploration or peer influences). The proportion of young adults affected by addictive behaviors further increases if non-substance-related or “behavioral” addictions, particularly internet-related disorders, are taken into account [9]. Specific data of the extent of internet-related disorders in the group of vocational students do not exist. However, as at least 36% of young people in Germany report problematic internet use, excessive media use seems to be highly likely in vocational students [9,10,11]. There are also links between non-substance and substance-related addictive behaviors in terms of risk factors and symptomatology [12], as is reflected in the introduction of the category “addictive behavior” in the International Classifications of Diseases 11 (ICD-11) [13].

Apprentices and students in vocational training therefore represent an important target group for prevention and health promotion measures. In Germany, there are two settings for the implementation of those measures, either in companies or in vocational schools. While the organizational environment is particularly suitable for structural preventive measures, it is often difficult for companies to offer behavior-oriented measures. This especially applies to small and medium-sized enterprises (SMEs), which train 80% to 90% of apprentices in Germany [14]. The vocational school setting is therefore highly relevant for behavior-oriented prevention efforts as it represents a setting where a large number of individuals can be reached with comparatively little effort in a development-oriented approach [15].

The majority of empirically evaluated school-based intervention programs is developed for regular school settings with younger target groups, while there has been a lack of comparable programs for vocational schools [16]. A 2016 meta-analysis summarized the results of 288 school-based drug prevention programs including half a million students and identified several effective age-appropriate intervention approaches [15]. Successful universal school-based programs for the prevention of alcohol and tobacco use in late adolescence (grades 10–12) used one or more of the following program elements: (1) promotion of self-control, (2) problem-solving techniques, (3) health education, (4) rejection skills and social influences, (5) social norms, (6) techniques of cognitive-behavioral therapy, and (7) involvement of peers and parents. Less is known on effective interventions for internet-related problems [17]. However, the basic mechanisms of the emergence and maintenance of internet-related disorders appear to be largely equivalent to the pattern of substance-related addictions, making it likely that the same intervention elements could be effective [18,19].

A popular school-based prevention approach is the voluntary commitment to abstain from or reduce habitual behaviors like substance use. For example, one of the most widespread programs for the prevention of smoking in secondary schools in Germany is the smoke-free class competition “Be Smart—Don’t Start”. The core of the program is a joint voluntary commitment of school classes to stay smoke-free for a period of 6 months. It focuses on influencing social norms, promoting self-regulation, and addressing social influences by deploying cognitive-behavioral intervention techniques [20]. This approach has been adapted to the prevention of binge drinking among older adolescents in the regular school setting and has been proven effective in a randomized study [21]. Further, the results of a recent controlled study in Germany suggested that a 20-min reduction of daily social media use over a period of 2 weeks is positively associated with well-being and a healthier lifestyle among students [22]. Specifically, the short-term reduction of social media use not only led to a lasting reduction of social media time and level of addictive symptoms, but also reduced the amount of daily smoked cigarettes over a period of three months.

To fill the gap in addiction prevention approaches for vocational students as a subpopulation with elevated risk for substance abuse and high media use, an app-based program has been developed, using a voluntary abstinence paradigm and addressing a wide selection of behaviors to incorporate a broad target group. The present study aimed to evaluate its effectiveness using a randomized design. It is expected that the intervention will increase the awareness of habitual behaviors (e.g., substance and digital media use) and will therefore lead to measurable reductions in these behaviors even after the end of the abstinence or reduction period. It is also expected that a temporary abstinence of a specific behavior will simultaneously affect other health behaviors and indicators of subjective well-being, even if the intervention does not directly address them.

## 2. Materials and Methods

### 2.1. Study Design

We conducted a two-arm multicenter, cluster-randomized, wait-list controlled trial with repeated measurements. Data were collected class-wise in schools at baseline and follow-up by using digital questionnaires and throughout the app-based intervention (only IG, CG did not use the app). The detailed study protocol is available online [23].

### 2.2. Participants, Recruitment, and Randomization

Schools were consecutively recruited via local authorities or direct contact using digital and printed information materials, school conferences, etc. After initial agreement from school principalities, research staff, social school workers, or principals contacted the teachers at the participating school and informed them about the study’s aims and procedures. Due to the COVID-19 pandemic, we faced unusual challenges regarding school recruitment and teacher involvement. Since enrolment, data assessment, and introduction of the app was planned to take place in the classroom, teachers were considered an important factor for the implementation and motivation of students. Recruitment occurred on class level, and teachers or principals decided if their classes were part of the study. Still, all students decided voluntarily for themselves. Students that were not involved with their class were not able to take part individually. In some instances, teachers enrolled multiple classes.

For randomization purposes, two classes were each paired into similar dyads based on three class characteristics: (1) frequency of in-school education, e.g., daily, twice a week, block lessons; (2) the educational area, e.g., technical/IT, services, and trade; and (3) year of training, e.g., first, second, third. Paired classes were then randomized into the Intervention Group (IG) or Control Group (CG).

We consecutively recruited 17 schools to participate in the study between August 2020 and December 2022. The teachers of 277 classes agreed to participate and 4591 vocational students out of these classes completed the baseline survey between 16 March 2021 and 26 April 2022. In total, 139 classes were assigned to the intervention group, 138 classes to the control group. The average number of participating students per class was 16.6 with a minimum of two and a maximum of 31. A total of 864 students were lost to follow-up due to absence at the follow-up assessment date and for 861 students, baseline and follow-up data could not be matched due to missing codes. Overall, 14 classes collectively withdrew from the study before follow-up. After follow-up assessment, 2861 students’ baseline and follow-up questionnaires remained for primary data results in an Intention-to-Treat (ITT) analysis. The average time between the two assessments was 7.7 weeks (SD = 3.08), with IG at 8.1 weeks and CG at 7.4 weeks. Figure 1 shows a detailed overview of the student numbers throughout the course of the study.

### 2.3. Intervention and Setting

“Meine Zeit ohne—Die Challenge“ (“My time off—the Challenge”) is an app-based intervention developed for implementation in vocational schools. It is based on the ideas of a project called “initiated abstinence” [24]. Intervention group students received an introduction to the overall theme of habits and on potentially risky health behaviors concerning substance and media use as well as gambling. Introductions were conducted by their teachers or—if not feasible—by research staff and lasted from 10 to 30 min depending on the time provided by schools and teachers. It was supported by a short explanation video demonstrating the use and goals of the app. The 1-min video summarized the selectable behavior areas, the functionality of the app, and all tasks available in the application. Students were then invited to install the app on their smartphones and to choose a meaningful challenge of behavior change. They could either abstain from or reduce their behavior for 14 days in the areas of smoking (cigarettes/e-products), alcohol, cannabis, digital media use (social media, video gaming, or streaming), gambling, or “another habit”. If the latter option was chosen, students were asked to define an abstinence goal. For the challenge areas alcohol, gambling, and cannabis use, students could only abstain from the behavior. The research staff or teachers did not check whether students started the app or challenge but helped with technical issues if necessary. After choosing an area of behavior change and starting the challenge, the participants received daily push notifications to assess their confidence in maintaining their goal for the next 24 h. In addition, they were asked whether they had been successful in pursuing their goal on the previous day. Push notifications were automatically sent in the morning and at noon. Following the last rating on the 14th day, the students were able to download and share a certificate of completion regardless of their results concerning the challenge. Students in the control group received no treatment but were provided with access to the app after completion of the follow-up survey.

### 2.4. Data Collection and Outcomes

Participation in the survey and intervention (IG) was voluntary, so the number of participating students could differ noticeably from previously announced class sizes. There were no exclusion criteria. After interested teachers were identified at a respective school, assessment/introduction dates were set for each class individually. Due to the COVID-19 pandemic, some introductions and survey assessments were conducted via video chat projecting a research staff member onto a classroom monitor. All students received a random code on paper, which granted them access to the digital questionnaire. Codes were handed out by research staff members directly before data assessment. In cases where physical presence by research staff was not permitted, codes were sent to teachers beforehand. Research staff helped with technical difficulties or language issues. IG participants’ codes also served as login codes for the intervention app. All students were encouraged to take a smartphone picture of the code and keep their codes to gain access to the follow-up questionnaire. To provide optimal data protection, this code was the only possibility to match data of baseline and follow-up questionnaires as well as app usage data. Research staff also attended the follow-up data collection in presence or online.

### 2.5. Measurements

Sociodemographic data included age, gender, income, parent’s migration history (yes/no), income, highest previous education, educational area, and educational progress. Primary outcomes were substance-related behavior in the last 30 days using binary answers (yes/no) as well as quantitative instruments. The assessment of alcohol use patterns was based on Audit-C assessing frequency and quantity of both drinking and binge drinking [11]. For cigarette, e-cigarette, and other e-product usage, the number of days and the number of cigarettes or appropriate units of e-products (15 puffs or 10-min usage) were assessed. Additional primary data were social media usage, gaming time, and gambling. Quantitative measurements for social media use and gaming included the number of days in the last month as well as overall used hours and minutes. Gambling was assessed dichotomously (yes/no). Secondary outcomes included physical activity (past month frequency), positive mental health, and general self-efficacy (short scale, three items) [25]. An extended description and overview of all measurements can be found in the study protocol [23].

For the present analysis, we created dichotomous health-promoting change variables (yes/no = 1/0) for all outcomes. For alcohol, gambling, and cannabis, a health-promoting change was defined as a positive 30-day prevalence at baseline vs. a zero 30-day prevalence at follow-up. For the use of cigarettes and e-products, change was defined as a reduction of at least 50% in the monthly number of cigarettes or e-product units from baseline to follow-up [26]. For social media and gaming, change was defined as a reduction of daily screen time by at least 20 min [22]. As an additional primary outcome, we created a new variable called General Adverse Health Behavior Improvement (GAHBI). This variable was coded 1 (yes), if one of the variables described above was positive, meaning that there had been at least one health-promoting change in any of the areas. Likewise, for the secondary outcomes an elevated score at follow-up compared to baseline was coded 1, while a decrease or no change was coded 0. 

### 2.6. Statistical Analysis

Based on previous studies [6,13,21], we calculated the sample size to detect a minimum difference of 20% between groups with a supposed drop-out rate of 30% on student level and 80% power requiring 4500 students from 225 classes. This size was achieved, although drop-out was slightly higher (37.7%).

For descriptive baseline data, we used standard code to tabulate ratios for data on nominal and ordinal scale levels. Metric data are presented with means and standard deviations (SD). As part of the attrition analysis, linear regressions were applied to evaluate differences between students that completed the follow-up survey and students that dropped out of the study. Differences between groups for change of all dichotomous outcomes from baseline to follow-up were analyzed using multilevel logistic mixed-effects regression models to account for the clustered structure of the data, with clustering occurring within federal states, schools, and classes [26]. Fixed effects were group (IG vs. CG) and students’ highest education. Education was added as covariate as there were differences between the intervention and control groups at baseline. Random intercepts were added at federal state, school, and class level. Odds ratios, 95% confidence intervals, and p-values were calculated and are reported for all outcomes. Sensitivity analyses included 1) computing the models for all behaviors separately, using only the actual app-users of the respective challenge area, and b) an analysis with all individuals assessed at baseline. In this second analysis, missing data were imputed using the MICE technique (multivariate imputations by chained equations) with M = 100 imputations [27]. The pooling of the regression estimates followed Rubin’s rule [28]. All analyses were conducted with Stata 17.0 Standard Edition.

## 3. Results

The average age of students at baseline was 19.2 years, with 53.9% being male, 45.2% female, and 0.8% not identifying as binary. Approximately one third had a migration background on both parents’ family sides. Health behavior data showed a 30-day smoking prevalence of 35.1%, a 64.3% 30-day alcohol prevalence, and a 30-day cannabis use prevalence of 15.4%. Furthermore, social media was on average used for 216.3 min per day and students reported on average playing video games for 90.2 min on a daily basis. Table 1 shows all relevant baseline characteristics of demographic variables as well as primary and secondary data related to the assessed behavioral change areas as well as attrition analysis and baseline data of the analyzed sample—differentiated for IG and CG.

Attrition analysis shows several differences between drop-outs and completers (i.e., participants who provided matching baseline and follow-up data) at the socio-demographic level. Student drop-out was related to a higher age, non-male gender, parents’ migration background, lower monthly income, and low education. Regarding behavioral data, attrition was related to a higher 30-day smoking (cigarettes and e-products) and cannabis prevalence as well as a higher number of daily social media minutes. Conversely, attrition was also related to a lower 30-day alcohol prevalence. Lower self-efficacy, positive mental-health, and less physical activity were also associated with dropping out.

Among the 2861 students used for the outcome analysis (“completers”), participants in the IG and CG had a statistically significant difference concerning the proportion of students with a high school diploma (as their highest previous education) (in favor of IG). This was accounted for in all further analyses.

Table 2 displays the descriptive results for the primary and secondary outcomes by showing the total number and proportion of health-promoting changes observed in the intervention and control groups. It also presents the number of app challenges started in each behavioral area.

Of the 1458 students in intervention classes who provided complete data, about half (50.4%, *n* = 735) used the app and started a 2-week abstinence challenge. As a challenge area, social media was selected by 332 students, alcohol was selected by 128, and smoking/vaping by 127 participants. Gaming, cannabis, and gambling were chosen by 79, 38, and 31 students, respectively. At the follow-up assessment, behavioral changes were observed in the area of social media use, with 662 (45.4%) students in the intervention group reporting a reduction of social media time by 20 or more minutes per day. In the control group, 563 students (40.1%) reported such a reduction at follow-up. For playing video games, there was also a high number of students who reported a behavioral change (25.6% in the IG vs. 24.0% in the CG). The number of post-intervention changes in the remaining areas were (in declining order): cigarettes, e-products, alcohol, gambling, and cannabis. Overall, a cumulated 1036 (71.1%) students in the intervention group reported any such change, compared to 946 (67.4%) in the control group (newly created GAHBI variable).

The multi-level mixed-effect regression models estimated an odds ratio (OR) of 1.31 (95% confidence interval 1.05–1.46) in favor of the intervention group to change social media behavior, which was the only statistically significant difference between groups. The OR for playing video games was 1.09 (0.91–1.31), for alcohol 1.12 (0.84–1.50), for gambling 1.14 (0.81–1.61), for cannabis 1.08 (0.73–1.59), for e-products 1.12 (0.85–1.47), and for cigarettes 1.12 (0.86–1.44)—all in favor of the intervention group. Added up, this resulted in a more than 20% higher chance (OR = 1.24, CI = 1.05–1.46) for the students in the intervention group to experience any health-promoting behavior-related change (GAHBI). Figure 2 presents the ORs and 95% confidence intervals of all primary and secondary outcomes. ORs for changes in the secondary outcomes were statistically not significant (general self-efficacy: OR = 1.05, CI =0.9–1.22, physical activity: OR = 1.05, CI = 0.86–1.27; positive mental health: OR = 1.15, CI 0.99–1.34), see Table 3.

### Sensitivity Analysis

In addition to the modified ITT-analysis, we conducted a sensitivity analysis including only the sub-sample of actual app users (i.e., those participants in the IG who at least started the challenge). In this analysis, each regression model’s independent group variable only included the intervention group student sample that had actually chosen this respective behavior for their challenge (e.g., the social media reduction analysis only included the 332 students that chose social media as their 14-day challenge).

Table 3 shows the results of the original ITT regression models compared to the results of the sensitivity analysis. In these, the OR of the intervention group compared to the control group was 1.79 (CI = 1.40–2.30) for the change in social media time and 2.18 (CI = 1.36–3.47) for the change in time of playing video games. Participants that engaged in a smoking-related challenge had a more than three times higher frequency of reduced smoking (OR = 3.27, CI = 2.01–5.31) and more than twice the frequency of a reduction in the use of e-products (OR = 2.50, CI = 1.46–4.29).

The sensitivity analysis with multiple imputed data using the full sample of 4591 students showed no meaningful differences compared to the complete cases analysis.

## 4. Discussion

Our study shows first indications of a positive effect of the app-based intervention in terms of an overall improvement of health-related behavior among vocational students.

With that, the study assessed behavior changes that exceeded the 2-week period of the challenge. It should be stressed that students were not encouraged to abstain from or reduce any behavior further than the period of the challenge, and the observed longer behavior changes can thus be assumed to be completely intrinsically motivated. Two driving forces, prompted through participation in the IG, can be assumed to underly this continuing behavior change; first, an elevated awareness for problematic use of substance or media, and second, techniques which were applied during the challenge.

On a general level, this study replicates the finding that school-based interventions that encourage students to commit themselves to abstain from unhealthy behaviors can lead to lasting future changes in health behavior [15]. Up to now, little is known about effective interventions in this older age group, especially in the group of vocational students, who seem to be at elevated risk. The present study contributes to this knowledge gap by transferring and adapting an established behavior-oriented preventive approach to the vocational school setting [23]. The results are promising as they show that small intervention effects are attainable in a setting with participants already having had years of experience with substances as opposed to primary prevention for adolescents with no or less experience. Changing habits is a learning process [29], which is directly prompted by the participation in the app-based challenge under study. However, long term habit change is unlikely to be achieved by one prevention intervention alone [30]. Ideally, interventions are received continuously over the life span, and are set in different environments of relevance [31]. We developed a promising looking stepping stone for school-based prevention for the target group of vocational students that lack targeted addiction prevention programs.

One new feature was the digital format and exclusive app control of the intervention. eHealth interventions targeting multiple behaviors offer increased student engagement, fidelity, and scalability for initially face-to-face based approaches. This facilitated reach of the target population and enabled individual challenges that were independent from the school context and teacher responsibility as is considered important [23]. The intervention also provides a low-cost service that could easily be disseminated across vocational schools later on. A second feature was the broad scope of the intervention. While previous studies mostly focused on specific substances, the voluntary commitment included a reduction or complete temporary abstinence from different substances (alcohol, nicotine, cannabis) and included non-substance-related behaviors like gambling, gaming, streaming, or social media use. This has the advantage of a high “reach”, as almost every student is given the opportunity to find a personally relevant challenge area. This narrows the gap between universal and selective prevention, as a universally applicable tool with selective areas. However, a possible disadvantage of this approach is that students might not choose areas they consider most important for themselves, but areas they consider the easiest or least adverse. As could be seen in the data, the majority of students chose the social media challenge, which is in line with the very high prevalence of social media and messenger use in the target population. But there were also high rates of alcohol, tobacco, and cannabis use in the analyzed sample, and only few students chose these challenge areas. Hence, further analyses will need to take a closer look into the factors influencing specific patterns of app usage and whether this has an influence on intervention effects.

Another empirical question relates to the size of the found intervention effects, i.e., their practical significance. The absolute difference between the two study groups must be seen as rather small (6.5% absolute difference for social media reduction, 3.7% absolute difference for GAHBI). Relevance in this context has two aspects, one on public health and one on the individual level. Following the principles of the prevention paradox, even small individual effects on health behavior have a large impact on public health when the intervention is applied on a large scale and therefore reaches many individuals. 

From an individual perspective, the chosen cut-points for change in the measured indicators seem meaningful, too. For example, evidence by Brailovskaia and colleagues indicated that a daily 20-min reduction of the use of Facebook can have lasting positive effects on Facebook involvement, on psychological well-being as well as healthy lifestyle [22]. Given the latter finding, it was expected that a reduction or abstinence of a specific behavior or substance will simultaneously affect other behavioral areas as well. While the main analysis of the present study indeed found a systematic tendency for students in the intervention group to have more favorable changes in all assessed outcomes, it was unable to replicate this healthy lifestyle transference effect, as none of these changes were statistically significant. More specifically, in line with a more specific route of influence, sensitivity analyses revealed that changes in smoking behavior were found for those who chose a smoking challenge and reductions in gaming times were found for those who specifically chose the gaming challenge. The latter results are, however, based on small sample sizes and therefore need to be replicated in future studies. When interpreting the small effect sizes, within and beyond the behaviors chosen in the challenge, the outcome measures have to be taken into account. They were chosen in correspondence to the behavior during the challenge, i.e., complete abstention with regards to alcohol, cannabis, and gambling. A continuous measure would have been more sensitive to reductions without full abstinence. Research in regular schools indicates that among older students, alcohol prevention is likely to have an effect on frequency of drinking rather than drinking rates [32].

### Limitations

This study has a number of limitations. First, randomization on class level reduced the overall clusters to 277. Despite efforts like pairing randomization units by similar characteristics, differences between the IG and CG participants were found, leading to possible selection bias. This has been taken into account by adding the respective variable as a covariate into the outcome analyses. Second, the attrition rate is high, which was partly caused by the anonymization procedure, allowing a link between baseline and follow-up data without collecting any personal information. While about 19% of participating students did not take part due to absence, another 19% could not be included into the main analysis because of a missing code and thus missing link to baseline data. Third, attrition analysis showed a number of highly significant differences between drop-outs and completers, which reduces the external validity of the study. However, we found no evidence to suggest that this affected the interpretation of findings across groups. Fourth, deviant from the study protocol we did not report primary outcomes as metric scores but as dichotomous changes across all or within one behavioral area. This procedure helps illustrating the results but reduces variance in the outcomes. To ensure that the presented results are not solely due to the dichotomization, the regression models were repeated using continuous outcomes and comparable results were found. For example, there was also a significant difference between the IG and CG groups in the absolute minutes used for social media. Fifth, it was not possible to study longer-term effects of the intervention as the observational period was too short. Lastly, the COVID-19 pandemic and associated conditions for schools and teachers clearly impacted the recruitment process. The pandemic—among others—was a regular argument for schools to not fully participate in the study. Digital components like the app itself or video calls helped greatly if the presence of research staff was not allowed because of contact restrictions. However, the recruitment might have been significantly easier and less selective in non-pandemic times.

## 5. Conclusions

Substance use and digital media exposure are major health risks for adolescents and young adults beyond secondary school. At the same time, they form habits over time, which are already present at young adulthood. This is the first time that a prevention program for vocational schools was rigorously evaluated in a randomized study in Germany. Although the recruited sample cannot be considered representative of vocational schools in Germany, the generalizability of the results is facilitated by the inclusion of a broad range of vocational sectors (i.e., service industries, business, and administrative professions as well as industrial-technical professions) within schools from three different states across Germany. The “Meine Zeit ohne” intervention was found to have promising results at feasibility for the target group and small but significant effects regarding students’ health behavior. In particular, participants choosing to reduce a specific substance- or media-related behavior seem to be successful at reducing this behavior for a short term. Implications for future research are the need for longer-term follow-up and to examine participant characteristics and the intervention mechanisms’ impact on app usage as well as effectiveness. The study also indicates that a large number of adolescents and young adults can be reached using a digital, low-threshold application, which can be used independently from the school or work context.

## Figures and Tables

**Figure 1 ijerph-20-01970-f001:**
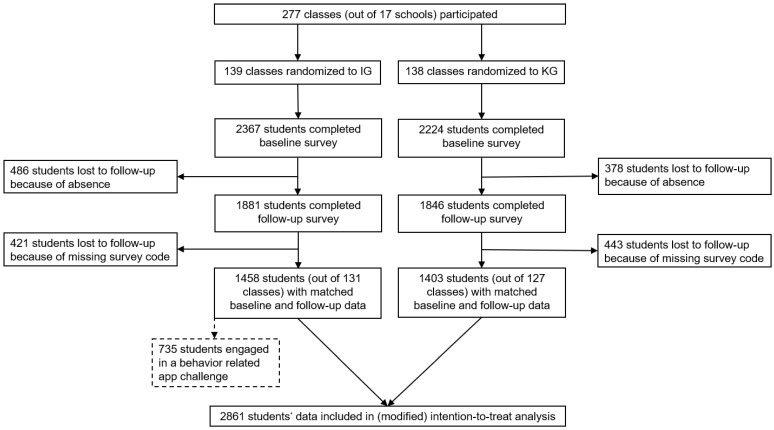
Participants flow chart.

**Figure 2 ijerph-20-01970-f002:**
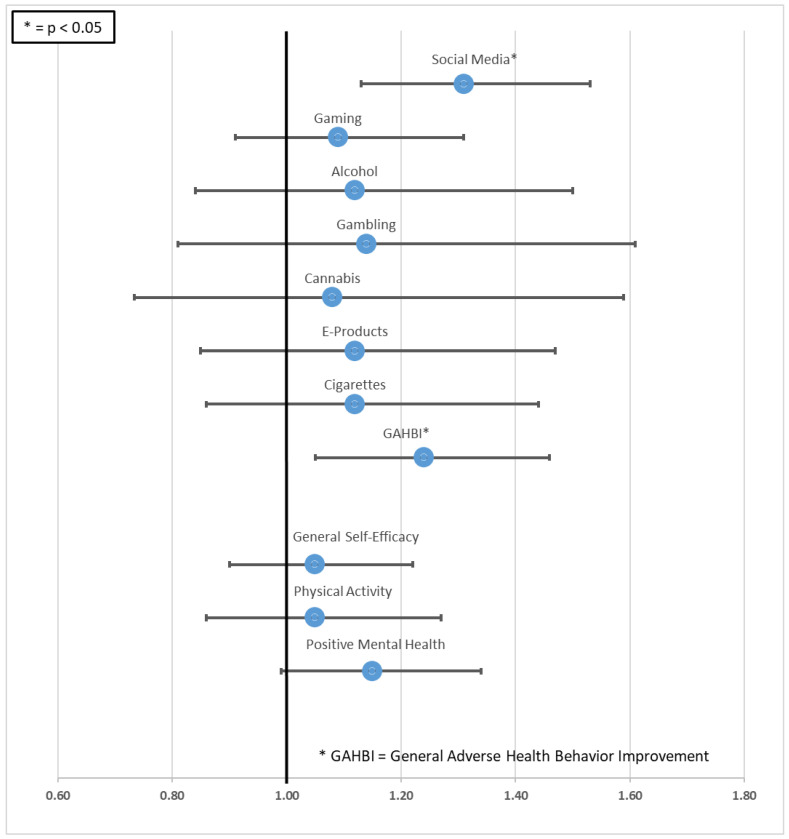
Change in IG participants’ behavior compared to the control group (odds ratios adjusted for highest education with 95% CI).

**Table 1 ijerph-20-01970-t001:** Baseline statistics, attrition analysis, and differences between the intervention and control groups.

	Full Sample—Attrition Analysis				Completer		
	Overall Sample	Drop-Outs	Completer	*p*-Value	Intervention	Control	*p-*Value
Students	4591	1730	2861		1458	1403	
**Socio-demographics**							
Age, years, mean (SD)	19.2 (4.2)	19.5 (4.2)	19.0 (3.7)	< 0.001	19.0 (3.5)	19.1 (4.38)	0.303
Gender							
- male	53.93%	50.29%	56.13%	< 0.001	56,93%	55.31%	0.351
- female	45.24%	48.79%	43.10%		42.39%	43.83%	
- other	0.83%	0.92%	0.77%		0.69%	0.86%	
Migration background							
- No	66.00%	56.71%	71.62%	< 0.001	72.22%	70.99%	0.465
- Yes	34.00%	43.29%	28.38%		27.78%	29.01%	
Monthly Income (€)							
> 1000	14.81%	15.02%	14.68%	< 0.001	12.97%	16.46%	0.862
600–999	47.37%	42.21%	50.51%		54.18%	46.69%	
< 600	37.81%	42.77%	34.81%		32.85%	36.85%	
Highest Education							
High School Diploma	17.62%	17.92%	17.44%	< 0.001	20.03%	14.75%	< 0.001
Middle School Diploma	53.57%	46.82%	57.64%		57.40%	57.87%	
Below Middle School Diploma	28.82%	35.26%	24.92%		22.57%	27.37%	
Educational area							
Vocational preparation	5.31%	8.67%	3.29%	0.005	3.16%	3.42%	0.177
Commerce, Industry and Technology	55.72%	45.78%	61.73%		60.77%	62.72%	
Economics and management	22.83%	28.27%	19.54%		19.89%	19.17%	
General school-based education	16.14%	17.28%	15.45%		16.19%	14.68%	
**Health behavior and subjective well-being**							
Smoking, 30 day prevalence	35.09%	37.80%	33.45%	0.003	31.89%	35.07%	0.072
E-products, 30 day prevalence	23.48%	26.36%	21.74%	< 0.001	22.22%	21.24%	0.524
Alcohol, 30 day prevalence	64.27%	58.15%	67.98%	0.008	67.90%	68.07%	0.924
Cannabis, 30 day prevalence	15.42%	18.27%	13.70%	< 0.001	14.13%	13.26%	0.498
Gaming, minutes/day, mean, SD	90.2 (114.3)	86.2 (115.6)	92.6 (113.3)	0.065	92.8 (112.0)	92.5 (114.8)	0.947
Social Media, minutes/day, mean, SD	216.3 (148.6)	232.5 (161.2)	202.7 (129.58)	< 0.001	199.2 (128.4)	206.3 (130.8)	0.143
Gambling, 30 day prevalence	12.22%	12.14%	12.27%	0.897	12.69%	11.83%	0.485
General Self-Efficacy, mean, SD	3.7 (0.8)	3.6 (0.8)	3.7 (0.8)	< 0.001	3.7 (0.7)	3.7 (0.7)	0.887
Positive Mental Health Scale, mean, SD	17.5 (5.9)	16.8 (6.1)	17.8 (5.8)	< 0.001	17.9 (5.7)	17.8 (5.9)	0.736
Physical Activity, frequency				0.016			0.564
- Never	8.43%	9.94%	7.51%		7.82%	7.20%	
- Less than once per week	34.57%	35.09%	34.25%		34.29%	34.22%	
- More than once per week	57.00%	54.97%	58.24%		57.89%	58.58%	

**Table 2 ijerph-20-01970-t002:** Assessed behavioral and well-being variables, criteria on which a health-promoting change from baseline to follow-up was defined, distribution of these changes in the IG and CG, and challenges started in each area (IG only).

Health Behavior	Health-Promoting Changes	Changes in IG (%)	Changes in CG (%)	Challenges (N)
Social Media	Screen time reduction ≥ 20 min	705 (48.35%)	587 (41.8%)	332
Gaming	Screen time reduction ≥ 20 min	390 (26.8%)	354 (25.2%)	79
Alcohol	30 day prevalence Yes to No	126 (8.6%)	112 (8.0%)	128
Gambling	30 day prevalence Yes to No	98 (6.7%)	84 (6.0%)	31
Cannabis	30 day prevalence Yes to No	66 (4.5%)	58 (4.1%)	38
E-products	Puffs down by at least 50 % **	144 (9.9%)	131(9.3%)	127 ^α^
Cigarettes	Cigarettes down by at least 50 % **	159 (10.9%)	140 (10.0%)	
GAHBI *	Any one of the changes above	1036 (71.1%)	946(67.4%)	735
**Subjective Well-Being**				
General Self-Efficacy	Elevated Score	505 (34.6%)	474 (34.6%)	
Physical Activity	Elevated Score	324 (22.2%)	298 (21.2%)	
Positive Mental Health	Elevated Score	621 (42.6%)	551 (39.3%)	

* = General Adverse Health Behavior Improvement; ** = Monthly; α = For cigarettes and e-products technically the same challenge had to be selected.

**Table 3 ijerph-20-01970-t003:** Odds ratio with 95% CI for intervention group behavior change compared to the control group—full IG sample and sensitivity analysis including only participants who chose a challenge in the respective health behavior area.

Full IG Sample (*n* = 1458)				Only Students That Chose the Respective Outcome			
Health Behavior	OR	95 %–CI	*p*-Value	Number	OR	95%–CI	*p*-Value
Social media	1.31	1.13–1.53	<0.001	332	1.79	1.40–2.30	<0.001
Gaming	1.09	0.91–1.31	0.299	79	2.18	1.36–3.47	0.001
Alcohol	1.12	0.84–1.50	0.447	128	0.92	0.46–1.84	0.803
Gambling	1.14	0.81–1.61	0.453	31	2.60	0.86–7.91	0.091
Cigarettes	1.12	0.86–1.44	0.402	127 ^α^	3.27	2.01–5.31	<0.001
E-products	1.12	0.85–1.47	0.407		2.50	1.46–4.29	0.001
Cannabis	1.08	0.73–1.59	0.697	38	1.91	0.56–6.44	0.300
GAHBI *	1.24	1.05–1.46	0.010				
**Subjective Well-Being**							
General self-efficacy	1.05	0.90–1.22	0.553				
Physical activity	1.05	0.86–1.27	0.640				
Positive mental health	1.15	0.99–1.34	0.068				

* = General Adverse Health Behavior Variable; α = For cigarettes and e-products technically the same challenge had to be selected.

## Data Availability

The study material (information sheets) is available to the public and can be found on the following website http://www.meine-zeit-ohne.de and by request. The MZo application is suitable for iOS (iOS 11.0 or newer) and Android (version 6 or newer) and is available in the app store for iOS devices and Play Store for Android devices. Anonymized study data and statistical codes for the analysis can be made available on request.

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
