# Peer review of "Effects of an App-Based Intervention Program to Reduce Substance Use, Gambling, and Digital Media Use in Adolescents and Young Adults: A Multicenter, Cluster-Randomized Controlled Trial in Vocational Schools in Germany"

_ijerph, 2023, doi:10.3390/ijerph20031970_

Round 1

Reviewer 1 Report

Great Job and Congratulations!

However, I have a few suggestions for your consideration:

1. Kindly provide a detailed explanation in the introduction of why did you bring both types of addiction together.

2. One implication section could be helpful in improving your paper.

All the best.

Reviewer 2 Report

I propose rewording the title of the paper so that it is clear what substances are meant (line 3). At the moment, one can only guess. The same remark applies to the Introduction section (line 34) and the abstract section (line 16: "problematic substance use").

Please check your work for punctuation errors.

Introduction.

Lines: 34, 40. Please see the comment above.

In the Introduction section, the authors should explain in detail what substances are involved and list them.

Line 38: How old are vocational school students? It might be worth adding this information. How long does vocational school education last?

Line 44: "other drugs" - please explain.

Line 84. Delete [.].

Line: 107. Please explain the abbreviations (IG, CG).

Lines: 115-117. The fragment needs to be explained. Please edit.

Lines: 118-123. Part is completely unclear.

Lines: 127-128. What does “….received a brief introduction….” mean?

Lines: 130-131. “It was supported by a short explanation video demonstrating……”? Explain? To expand?

The research methodology needs to be supplemented.

Was the study voluntary? Did the parents have to consent to the study? (The age of the students is unknown.) Were the test results anonymous? Part of the information is contained in the next point (2.4.). Some of the information may be transferred to point 2.2., so there are no doubts about the research methodology.

All data on the research methodology, number of participants, and study time should be transferred to the Materials and Methods section. Therefore, the number of schools and students participating in the study should be moved to the research methodology and not presented in the Results section. After all, the number of participants was not the subject of the study.

Lines: 210-236 Move the fragment to the Materials and methods section.

Line: 357. Was this analysis described in materials and methods?

Discussion.

Lines: 386-390. According to the reviewer, there is no need to repeat the purpose of the work.

Lines: 391-404. Isn't this excerpt a summary of the results?

Lines: 405-415. Shouldn't this fragment be moved to the Introduction section?

I am asking for a thorough, careful analysis of the entire work. I propose to consider the discussion of the results. It is best to expand it because there is little to be gained from it in its current form. Especially since part of it should be in the Introduction section.

Round 2

Reviewer 2 Report

Thanks to the authors for the changes made.

The work has been much improved.